# Effect of AMF Inoculation on Reducing Excessive Fertilizer Use

**DOI:** 10.3390/microorganisms12081550

**Published:** 2024-07-29

**Authors:** Siru Qian, Ying Xu, Yifei Zhang, Xue Wang, Ximei Niu, Ping Wang

**Affiliations:** 1State Environmental Protection Key Laboratory of Wetland Ecology and Vegetation Restoration, School of Environment, Northeast Normal University, Changchun 130024, China; qiansr464@nenu.edu.cn (S.Q.); xuying222@nenu.edu.cn (Y.X.); wangx881@nenu.edu.cn (X.W.); niuxm472@nenu.edu.cn (X.N.); 2Jilin Provincial Academy of Forestry Sciences, Changchun 130033, China; yifeii@hotmail.com

**Keywords:** mycorrhizal effect, phosphorus supplementation, plant trait, soil mycelium density, low-phosphorus soils

## Abstract

Excessive use of chemical fertilizer is a global concern. Arbuscular mycorrhizal fungi (AMF) are considered a potential solution due to their symbiotic association with crops. This study assessed AMF’s effects on maize yield, fertilizer efficiency, plant traits, and soil nutrients under different reduced-fertilizer regimes in medium–low fertility fields. We found that phosphorus supplementation after a 30% fertilizer reduction enhanced AMF’s positive impact on grain yield, increasing it by 3.47% with pure chemical fertilizers and 6.65% with mixed fertilizers. The AMF inoculation did not significantly affect the nitrogen and phosphorus fertilizer use efficiency, but significantly increased root colonization and soil mycelium density. Mixed fertilizer treatments with phosphorus supplementation after fertilizer reduction showed greater mycorrhizal effects on plant traits and soil nutrient contents compared to chemical fertilizer treatments. This study highlights that AMF inoculation, closely linked to fertilization regimes, can effectively reduce fertilizer use while sustaining or enhancing maize yields.

## 1. Introduction

Excessive use of chemical fertilizers has become one of the global environmental issues [1], which not only leads to nutrient loss [2] but also results in soil health problems in agricultural fields [3]. Reducing amounts of chemical fertilizers [4] or substituting some of them with organic or microbial fertilizers [5] are common solutions for the issue of excessive fertilizer use. However, improper use of these methods may result in crop yield reduction [6]. In recent years, the mutually beneficial symbiotic relationship between arbuscular mycorrhizal fungi (AMF) and crops has attracted more attention. AMF can establish a symbiotic relationship with most crops, promoting soil nutrient absorption by crops through the extensive mycelium and mycorrhizal networks, and enhancing fertilizer utilization efficiency. Therefore, reducing fertilizer inputs while applying AMF inoculants can maintain or even increase crop yields [7]. However, other studies did not find a positive effect of AMF on crop yield [8]. We suggest that the inconsistent findings in the current studies may be closely related to their different local fertilization regimes and background soil fertility levels.

The impact of AMF on symbiotic plants depends on the supply and demand of available nutrients needed by plants [9]. When the nutrients required for plant growth not only depend on root absorption directly but also indirectly through the mycelium network, AMF exhibits positive mycorrhizal effects. However, when the nutrients acquired by roots fully meet the plant’s requirements, the mycorrhizal effect of AMF may either be negative or negligible [10]. Therefore, determining appropriate levels of fertilizer input is crucial for AMF to help maintain crop yields with reduced fertilizer applications. The influence of fertilizer inputs on AMF mycorrhizal effects also depends on the nutrient release rate of the fertilizer [11]. Compared to organic fertilizers, the nutrients in chemical fertilizers are released more rapidly, which may suppress the activity of arbuscular mycorrhizal fungi, thereby reducing the potential for positive mycorrhizal effects on crops [12]. Phosphorus is essential for the growth and reproduction of AMF and plants [13]. Reducing the inputs of phosphorus fertilizer may inhibit AMF colonization and yield accumulation, especially in croplands where available phosphorus is commonly limited. Since most chemical fertilizers are nitrogen–phosphorus–potassium compounds, a reduction in nitrogen fertilizer inputs also results in a corresponding reduction in phosphorus and potassium fertilizer inputs. Therefore, when reducing the amount of compound fertilizer applied, it is also important to consider whether the amount of phosphorus input can meet the mycorrhizal crop’s demand for phosphorus.

Based on the above, doubts have been raised regarding the efficacy of AMF inoculation on crop yields, because after reducing fertilization levels, AMF inoculation has been reported to either decrease [14], increase, or maintain crop yields [15]. Therefore, we believe it is necessary to evaluate the mycorrhizal effects of AMF under different reduced fertilization regimes including factors such as fertilization levels, fertilizer types, and phosphorus levels. This study was conducted in maize croplands. Our hypotheses were: (1) the effect of AMF on maize yield in medium–low fertility croplands increases with reduced fertilization, and phosphorus supplementation, because (2) fertilizer reduction and/or phosphorus supplementation may promote the establishment of symbiosis between plants and AMF, (3) thereby generating a series of mycorrhizal effects in plants and soil, which were beneficial for maintaining or increasing crop yields. We aim for this study to contribute to a more comprehensive understanding of the role of arbuscular mycorrhizal fungi in practical on-farm fertilizer reduction, as well as its response to crucial fertilizer factors.

## 2. Materials and Methods

### 2.1. The Study Area

This study was conducted in Nong’an County, Changchun City, Jilin Province, China in 2022, located in the corn belt of China (44°10′17″ N, 125°2′24″ E). The climate in this region belongs to a temperate monsoon climate. Based on meteorological data from the Nong’an Meteorological Station (44°13′48″ N, 125°5′24″ E) spanning from 1951 to 2018, the average annual temperature in this area is 5.5 °C, with an accumulated temperature above 10 °C totaling 2880 °C, and a frost-free period lasting 143 days. The annual precipitation measures 516.9 mm, predominantly occurring from June to August, with an average annual evaporation rate of 1501 mm. The soil type in the area is black soil. In most maize fields in the area, chemical compound fertilizers are applied, with only a small portion of maize fields utilizing a combination of chemical and organic fertilizers [16]. To achieve high yields, the application rate of chemical compound fertilizers is generally as high as 260 kg Nha^−1^, which is higher than the recommended rates [17]. To save labor, all fertilizers are applied as basal fertilizers in a single application.

The experiment plots (32 m × 13 m) were established in maize cropland under continuous conventional management, where the excessive use of chemical fertilizers is more pronounced compared to organic agriculture. Soil samples were collected from the experimental field during spring 2022, and the following parameters were measured. The soil bulk density was 1.39 ± 0.07 g cm^−3^, the organic matter content was 3.31 ± 0.67%, the pH value in a soil–water slurry (soil/water: 1/5 *w*/*v*) was 8.50 ± 0.02, and the electrical conductivity of soil solution (soil/water: 1/5 *w*/*v*) was 188.25 ± 27.23 μs cm^−1^. The total nitrogen and phosphorus contents in the soil were 1.28 ± 0.03 g kg^−1^ and 0.33 ± 0.01 g kg^−1^, respectively, while the available nitrogen and phosphorus contents were 109.22 ± 10.85 mg kg^−1^ and 3.54 ± 0.62 mg kg^−1^, respectively. AMF is naturally and widely present in cropland soil [16], and the natural spore density of AMF in the cropland was detected as 23 ± 2.49 per 10 g of dry soil. The background soil fertility level in the experimental cropland is medium–low, especially with the available phosphorus content being the lowest among croplands [18].

### 2.2. Materials of Plant, Fertilizers and AMF Inoculant 

The maize variety used in the experiment is a commercial hybrid with the trade name ‘Yuhe 9’. The chemical fertilizer compound (N-P_2_O_5_-K_2_O = 26-11-13) was produced by Jilin Longyuan Chemical Fertilizer Co., Ltd., Changchun, China. The earthworm chelates organic fertilizer (N + P_2_O_5_ + K_2_O ≥ 8%, organic matter ≥ 20%) was produced by Beijing Juwo Biotechnology Co., Ltd., Beijing, China. Calcium superphosphate (P_2_O_5_, 16%) was used to adjust the phosphorus fertilizer application rate. Commercial AMF inoculants were not used in this study to avoid issues such as competition between foreign and local strains or the unsuitability of commercial strains for maize field soil environments [19,20]. Maize was used as the host plant to enrich and cultivate AMF from local maize croplands to obtain inoculants. The dominant AMF in the propagules belongs to the genera *Glomus* and *Claroideoglomus*, with a spore density of approximately 113 ± 11 spores per 10 g of dry soil (unpublished data).

### 2.3. Fertilizer Treatments

To assess the impact of different fertilization treatments on the efficacy of arbuscular mycorrhizal fungi in reduced fertilized agricultural fields, this study established six distinct fertilization regimes. These regimes encompassed both local traditional fertilization practices and a 30% reduction in fertilizer application, along with phosphorus supplementation after reduced fertilization under both chemical and mixed fertilizer regimes (Table 1). Each fertilization treatment was set up with and without AMF inoculation. Additionally, a control treatment without any fertilization and inoculation (Control) was set up to calculate fertilizer utilization efficiency. Mixed fertilization was applied following the guidelines of the earthworm chelate organic fertilizer, maintaining a weight ratio of compound chemical fertilizer to organic fertilizer at 2:1. 

The N input (182 kg ha^−1^) and the P_2_O_5_ input (77 kg ha^−1^) after a 30% reduction in fertilizer application, is similar to the recommended rate (N: 170–190 kg ha^−1^, P_2_O_5_: 70–80 kg ha^−1^) for maize in conventional croplands in Jilin Province [17]. Since the available phosphorus content of the background soil is very low, and phosphorus is susceptible to leaching or conversion into insoluble forms in alkaline soil [21], a phosphorus supplementation treatment was conducted. The phosphorus application rate under chemical fertilizer regime was adjusted to 110 kg P_2_O_5_ ha^−1^ through supplementation with calcium superphosphate, while under the mixed fertilizer regime, an equivalent amount of calcium superphosphate was applied then the phosphorus application rate was adjusted to 94 kg P_2_O_5_ ha^−1^. 

### 2.4. Planting and Inoculation

The planting was carried out in spring 2022. The fertilization treatments and inoculation treatments were arranged in a split-plot design in the experimental plots, with 6 fertilization treatments and one control as the main plots (randomly arranged on different ridges) and inoculation treatment as the sub-plots (randomly arranged along the direction of the ridges). Three buffer rows of maize were established between the plots with inoculation and without inoculation. The four replicate plots were also arranged in sequence along the direction of the ridges, with three buffer rows of maize between the plots. Excluding the buffer ridges and buffer rows, each plot of each treatment was planted with 10 maize plants, totaling 40 maize plants per treatment, and 560 plants in total (40 plants × 7 fertilizers × 2 AMFs).

The maize planting ridges were spaced at 65 cm intervals, with a plant-to-plant spacing of 25 cm. Following local fertilization practices, all fertilizers were applied in the planting furrows in a single application. Subsequently, the furrows were covered with soil to form ridges, and then watered and seeded. AMF inoculants (100 g per plant) were applied simultaneously with maize seeds, while the control treatment, devoid of inoculation, received an equivalent amount of sterilized AMF inoculant. The management practices during the maize growth period followed local conventional agriculture management, which included a single application of the herbicide ethofumesate for weed control. 

### 2.5. Sample Collection and Measurement Indicators

At the dent stage (R5) of maize growth, three maize plants were randomly selected and harvested from each plot of each treatment. Roots within a 30 × 30 × 30 cm^3^ area centered on the plant were carefully collected. Selected portions of the fine roots were randomly preserved in FAA fixative solution for assessing the frequency of mycorrhizal colonization, colonization intensity, and arbuscule abundance in maize roots. Due to its negligible weight, accounting for less than 1% of the total maize root biomass, this portion was excluded from root biomass calculations. After cleaning the roots, approximately 2 g (dried weight) of fine roots (<2 mm) were randomly selected and scanned (Epson V700 root scanner). Root length and average root diameter were subsequently analyzed using WinRHIZO Pro 2013e (Regent Instruments Inc., Quebec, QC, Canada). Specific root length (SRL) was calculated as the ratio of root length to root dry weight.

Each maize plant was partitioned into roots (including scanned roots), stems, leaves (including husks), and ears (including cob). Subsequently, these components were dried at 70 °C until a constant weight was attained and then weighed accordingly. The total biomass of each maize plant and the ratio of ear weight to total biomass were computed. Roots, stems and leaves, and grains were then ground separately and sieved through a 100-mesh sieve. The concentrations of total nitrogen were determined using an EA3100 elemental analyzer (Strada C.na Maestà, Pavia, Italy), while total phosphorus concentrations were measured using Mo-Sb spectrophotometry. The total nitrogen and phosphorus accumulated by each maize plant were calculated based on the weights of the respective organs.

Using a soil auger with an inner diameter of 40 mm, soil samples were collected at three randomly selected points within a 10 cm radius of the stem of each maize plant, at a depth of 0–20 cm. After naturally air-drying, the soil samples were ground and sieved through a 40-mesh sieve. Total carbon and total nitrogen concentrations in the soil were determined by elemental analysis (EA3100, EuroVector, Pavia, Italy), and organic matter content was calculated from the soil’s total carbon concentration, multiplied by a coefficient of 1.724 [22]. The available nitrogen concentration of the soil was determined by alkaline diffusion titration, and the total and available phosphorus concentration of the soil was determined by a continuous flow analyzer (SAN + +, SKALAR, Breda, The Netherlands) [23]. Phosphorus is extracted using a 0.5 M NaHCO_3_ solution adjusted at a pH of 8.5. Mycelium was extracted from the 0–20 cm soil layer using the membrane filtration method, and mycelium length was estimated using the gridline intersect method and converted to mycelium density [24]. Spores of AMF were extracted using the wet sieving-sucrose centrifugation method, and spore density was counted under a dissecting microscope [25,26]. The staining, slide preparation, and observation of root frequency of mycorrhiza, colonization intensity, and arbuscular abundance were conducted following the method of Trouvelot [27]. The related calculations were performed with INOQ Calculator Classic (INOQ GmbH, Schnega, Germany).

At the R6 stage of maize maturity, all plants were harvested, and the grains were dried and weighed to calculate the grain yield. Then, the agronomic use efficiency (AUE) and partial factor productivity (PFP) of nitrogen (phosphorus) fertilizers were computed. The formulas are as follows:(1)$$\mathrm{Agronomic}\;\mathrm{Use}\;\mathrm{Efficiency}\;=\frac{\left(Y-Y_{0}\right)}{F}$$
(2)$$\mathrm{Partial}\;\mathrm{Factor}\;\mathrm{Productivity}\;=\frac{Y}{F}$$
where Y and Y_0_ represent the grain yield in fertilized and unfertilized treatments, respectively, and F represents the amount of nitrogen (phosphorus) fertilizer applied. The unit symbols of AUE and PFP are kg kg^−1^.

### 2.6. Statistics and Data Analyses

First, linear mixed models were used to analyze the main effects and interaction effects of fertilization treatments and inoculation treatments on arbuscular mycorrhizal fungi (AMF) indicators (frequency of mycorrhiza, colonization intensity, and arbuscule abundance), grain yield, and 1000-grain weight, nitrogen (phosphorus) fertilizer use efficiency (AUE and PFP), soil mycelium density, and spore density. To determine whether significant differences existed among the fertilization treatments, the estimated marginal means obtained from the linear mixed models were used to conduct LSD tests for each dependent variable. Subsequently, within each fertilization treatment, independent t-tests were conducted for each dependent variable to detect whether significant differences existed between the inoculated and non-inoculated treatments.

To assess whether AMF inoculation significantly affected plant traits and soil nutrient content, the mycorrhizal effects on 13 plant traits and 4 soil nutrient indicators were calculated separately under each fertilization condition. The formula for mycorrhizal effects (ME) is as follows:(3)$$\mathrm{Mycorrhizal}\;\mathrm{effect}\;\left(\mathrm{ME}\right)=\frac{\left(X_{\text{AMF}}-X_{\text{non-AMF}}\right)}{X_{\text{non-AMF}}}$$
where X_AMF_ and X_non-AMF_ represent the indicators in inoculated and non-inoculated treatment, respectively.

Single-sample t-tests were performed on each plant and soil mycorrhizal effect to test whether the value significantly differed from 0. When the mycorrhizal effect value was significantly greater than 0, it indicated a positive mycorrhizal effect from AMF inoculation; when significantly less than 0, it indicated a negative mycorrhizal effect, and when equal to 0, it indicated no mycorrhizal effect.

Finally, a series of correlation analyses were conducted under inoculated and non-inoculated conditions, respectively, to explore whether AMF inoculation altered the correlation between the amount of nitrogen (phosphorus) applied and various indicators. These indicators included symbiotic mycorrhizae, yield and fertilizer utilization, plant traits, plant nutrient content, and soil nutrient content. This comprehensive assessment aimed to clarify how mycorrhizal inoculants contribute to reducing fertilizer use on farms. 

Statistical analysis was performed using R version 4.3.3, and graphs were generated using Excel for Microsoft 365 and the Chiplot platform [28].

## 3. Results

### 3.1. Frequency of Mycorrhiza, Colonization Intensity, and Arbuscule Abundance

Both fertilization and inoculation treatments significantly influenced the frequency of mycorrhizae, colonization intensity, and arbuscular abundance of roots. Additionally, the interaction between fertilization and inoculation treatments reached a significant level for the frequency of mycorrhizae (Table 2, Figure 1). Under both chemical fertilizer and mixed fertilizer conditions, reducing fertilizer application significantly increased the frequency of mycorrhizal colonization in maize roots, with nearly all maize roots being colonized by AMF (>98%). The enhancing effect of the inoculation treatment on the frequency of mycorrhizal colonization was dependent on the fertilizer dosage. 

Before fertilizer reduction, the colonization intensity of maize roots was 19.59 ± 2.08%. Following the reduction in chemical fertilizer application, both the colonization intensity and arbuscular abundance of maize roots significantly increased, and this effect was further amplified with phosphorus supplementation (Figure 1b,c). Similarly, after reducing mixed fertilizer application, the colonization intensity and arbuscular abundance of maize roots also significantly increased, although supplemental phosphorus fertilizer did not exert a significant effect on these two indicators (Figure 1b,c). Inoculation with AMF significantly increased the colonization intensity and arbuscular abundance of maize roots, albeit its enhancing effect did not achieve statistical significance in the RedChemP and Mix treatments (Table 2, Figure 1b,c).

### 3.2. Grain Yield

When applied with traditional chemical (Chem) or mixed fertilizers (Mix) application rate, the grain yields were 10.14 ± 0.49 t ha^−1^ and 9.66 ± 0.23 t ha^−1^, respectively. After reducing fertilizer application, grain yields in chemical (RedChem) and mixed fertilizers (RedMix) treatments decreased by 1.16% and 1.29%, respectively, compared to the grain yield of Chem. Phosphorus supplementation after fertilizer reduction led to an increase in grain yield (compared to Chem) by approximately 4.63% (RedChemP) and 6.38% (RedMixP), respectively (Figure 2a). 

Inoculating AMF under traditional chemical fertilizer conditions (Chem with AMF) led to a 6.75% decrease (compared to Chem) in grain yield but increased the grain yield by 3.47% when supplementary phosphorus fertilizer was added after reducing fertilizer application (RedChemP, Figure 2a). In the mixed fertilizer system, inoculating AMF increased the grain yield by 6.65% when supplementary phosphorus fertilizer was added after reducing fertilizer application (RedMixP, Figure 2a). The differences in 100 kernel weight among different fertilization and inoculation treatments did not reach a significant level (Figure 2b).

### 3.3. Fertilizer Use Efficiency

The influence of AMF inoculation on nitrogen (phosphorus) fertilizer use efficiency was not significant (Table 2). However, significant variances were observed in AUE of nitrogen and PFP of nitrogen and phosphorus across different fertilization treatments (Table 2, Figure 3). 

Reduced fertilizer application significantly increased the AUE of nitrogen (RedChem and RedMix increased by 38.15% and 37.92%, respectively), and enhanced the PFP of nitrogen (RedChem and RedMix increased by 41.20% and 41.89%, respectively). Supplementary phosphorus fertilizer further increased the AUE of nitrogen (RedChemP and RedMixP increased by 61.72% and 72.53%, respectively), and the PFP of nitrogen (RedChemP and RedMixP increased by 49.48% and 52.92%, respectively). Reduced fertilizer application did not significantly increase the AUE of phosphorus (Figure 3c), but significantly enhanced the PFP of phosphorus (Figure 3d, RedChem, and RedMix increased by 41.20% and 40.78%, respectively). 

### 3.4. Mycelium Density

Both fertilization treatments and AMF inoculation significantly influenced soil mycelium density (Table 2). In comparison to fertilization treatments, the impact of AMF inoculation on soil mycelium density in maize fields was more pronounced (Table 2), resulting in a significant increase in mycelium density in the soil (Figure 4a). When applied with mixed fertilizers, the impact of AMF inoculants on soil mycelium density was greater compared to the application with chemical fertilizers.

Significant variations in soil spore density were observed under different fertilization treatments (Table 2). Spore density exhibited similar trends under both chemical and mixed fertilizer applications, wherein reduced fertilizer application led to decreased spore density, and subsequent phosphorus supplementation after fertilizer reduction further decreased spore density (Figure 4b). The impact of AMF inoculation on soil spore density was not significant (Table 2). 

### 3.5. Mycorrhizal Effects on Plant Traits and Soil Nutrients

The mycorrhizal effects of AMF on plant traits and soil nutrients exhibited significant differences when applied with chemical fertilizer versus mixed fertilizer (Figure 5). In maize fields where pure chemical fertilizer was applied, most mycorrhizal effects of AMF on both plant traits and soil nutrients did not reach a significant level (Figure 5a–c). Conversely, in fields where mixed fertilizer was applied, several mycorrhizal effects of AMF reached significance (Figure 5d–f). Particularly when combined with reduced mixed fertilizer and phosphorus supplementation, AMF inoculation resulted in significant effects on maize morphological traits (increased root length), nutrient traits (increased leaf nitrogen concentration and root-to-leaf phosphorus concentration, decreased grain phosphorus concentration), and soil nutrient content (increased total nitrogen) (Figure 5f).

### 3.6. Correlation Analysis

Correlations of the nitrogen (phosphorus) input to plant root colonization, nutrient use efficiency, and some soil nutrient indicators were observed, regardless of whether AMF inoculants were applied (Figure 6). However, the correlations of the fertilizer input to plant traits reached significance only under inoculation conditions, and more correlations occurred with phosphorus input. Root colonization intensity and arbuscular abundance increased with reduced nitrogen application (Figure 6), while soil mycelium density increased with decreased nitrogen and phosphorus input (only under inoculation), indicating enhanced symbiotic mycorrhizal establishment with reduced fertilizer input. The correlations between nitrogen (phosphorus) fertilizer use efficiency and nitrogen (phosphorus) inputs were notably higher under inoculation treatment compared to non-inoculation treatment. Following AMF inoculation, the root–shoot ratio decreased, the spike weight ratio increased, and plant phosphorus content decreased with decreasing nitrogen (phosphorus) input, contributing to the maintenance or enhancement of grain yields. The relationship of nitrogen (phosphorus) input to soil nutrient content varied significantly between non-inoculated and inoculated treatments (Figure 6).

## 4. Discussion

### 4.1. Phosphorus Supplement after Fertilizer Reduction Benefits AMF to Maintain Crop Yield

The impact of AMF on maize yield was closely related to the fertilizer input. The traditional fertilization level in the study area far exceeded the demands of the crops, because the maize yield only decreased by an average of 1.23% after fertilizer reduction, and grain yield was uncorrelated with fertilizer input levels (Figure 2a and Figure 6). This excessive fertilizer input alters the interactions between plants and soil microbes [29], which decreases the maize yield when crops are inoculated with AMF. However, AMF inoculation (under mixed fertilizer application) with reduced fertilizer input or phosphorus supplementation after fertilizer reduction increased grain yield (Figure 2a). Therefore, the influence of fertilizer factors, such as application rate, phosphorus levels, etc., on mycorrhizal effects should be considered when applying AMF inoculants into agricultural fields. This result indicates that appropriate fertilizer regimes can enhance the positive impact of mycorrhizae on crops, potentially leading to improved yields, especially under reduced fertilizer application [30].

Additionally, the influence of AMF on crop yield could be influenced by variables such as prevailing soil conditions [31], duration of inoculation [6,32,33], and the choice of AMF strain [34]. Typically, AMF exhibits a more pronounced growth-promoting effect on plants in conditions of soil nutrient deficit or the presence of stress such as drought and salinity [29,35,36]. Consequently, employing AMF in low-fertility fields is preferable, as symbiotic mycorrhizae can facilitate fertilizer reduction without compromising crop yield [37,38]. For cropland subjected to prolonged excessive fertilizer application, reintroducing AMF to rebuild the mutualistic symbiotic relationship between crops and mycorrhizal fungi may take time for full restoration. This delay could be why this study, along with others [39], found that AMF inoculation improved crop yield, but the differences compared to non-inoculation treatments were not statistically significant. The choice of reintroduced AMF strains or species may also influence crop yield to some degree [40]. Local AMF strains are often more effective in enhancing crop yield compared to commercial inoculants because they are better adapted to local soil conditions and possess survival advantages [41,42]. However, commercial AMF inoculants are advantageous due to their efficient strains [43,44]. With careful selection or formulation of suitable strains and proportions for specific host plants and soil environments, commercial inoculants can also improve crop yield and quality while preserving soil ecosystem balance and stability [45,46].

Appropriately adjusting the nitrogen–phosphorus ratio could also serve as an effective strategy to mitigate excessive fertilizer usage [47]. In our experiment, the application of phosphorus supplementation after reducing fertilizer input resulted in an average increase in maize grain yield of 5.51% (Figure 2a). Furthermore, nitrogen fertilizer use efficiency and partial factor productivity showed significant improvements (Figure 3a,c), with a more pronounced promoting effect compared to AMF inoculation. This indicates that optimizing fertilizer nutrient ratios represents a viable approach to sustain or even enhance crop yield when fertilizer application is reduced. Given the significant influence of nutrient ratios on AMF and other microorganisms [48], further experiments are necessary to comprehensively evaluate the practical utility of adjusting fertilizer nutrient ratios in addressing excessive fertilizer use in farmlands.

### 4.2. Effects of Fertilizer Reduction and Phosphorus Supplementation on Establishment of Mycorrhizal Symbiosis and Mycorrhizal Effects 

The impact of AMF on plants depends on the mutualistic symbiosis established between both parties and their mycelium networks [49]. Typically, lower soil nutrient levels correspond to stronger root colonization by mycorrhizae [50]. Following the reduction in fertilizer application, whether with chemical or mixed fertilizers, the frequency, intensity, and arbuscular abundance of root colonization all significantly increased (Figure 1). Even in the absence of AMF inoculation, there was an upward trend in AMF root colonization following fertilizer reduction. This indicates the widespread occurrence of symbiotic relationships between crops and AMF in medium–low soil fertility farmlands. While continuous agricultural management disturbances may diminish the colonization ability of original AMF in farmlands [51,52], some effective propagules persist in the soil [53]. Reduced fertilizer application intensifies the dependence of crops on beneficial symbiotic fungi [54], consequently increasing root colonization (Figure 1). However, due to the limited colonization capacity of original AMF in farmland, a significant increase in colonization intensity, arbuscular abundance, and mycelium density of roots was observed only after additional inoculation with AMF inoculants which contained abundant spores and mycelium (Figure 1 and Figure 4). Moreover, mycelium density increased with decreasing nitrogen and phosphorus inputs (Figure 6). Hence, inoculating AMF after reducing fertilizer application significantly promoted the establishment of symbiotic relationships between crops and AMF, resulting in the formation of more mycorrhizal roots and mycelium.

Phosphorus, in addition to overall soil nutrient levels, plays a pivotal role in mycorrhizal formation and mycelial development [55]. Typically, as the phosphorus content in the soil increases, the colonization of plant roots by AMF tends to decline [13,56]. However, our findings indicated enhanced root colonization and mycelium density by the phosphorus supplementation following chemical fertilizer reduction, and no significant change following mixed fertilizer reduction (Figure 1 and Figure 4). In alkaline soil croplands, phosphorus in calcium superphosphate fertilizer is susceptible to leaching or conversion into insoluble forms [21], particularly when applied in pure chemical fertilizer regimes [57,58]. This finding suggested that phosphorus supplementation following reduced fertilizer application (or adjusting the nitrogen–phosphorus ratio) could promote AMF colonization of crop roots in conventional agricultural fields at low phosphorous levels. The microbial interactions and competition for nutrients in the soil ecosystem might have varied between the mixed and chemical fertilizer treatments, affecting the response of AMF inoculation to phosphorus supplementation. Further investigation into these interactions is necessary to fully understand the underlying mechanisms.

The phosphorus supplementation following mixed fertilizer reduction led to more pronounced mycorrhizal effects of the inoculated AMF on both plants and soil, as evidenced by significant mycorrhizal effects on specific root length, plant nitrogen, and phosphorus content, and total nitrogen content of soil (Figure 5). Arbuscular mycorrhizal plants altered their biomass allocation strategy to roots and other organs due to the ability of mycelium to absorb and transport nutrients and water [59]. For instance, with decreasing nitrogen (phosphorus) input application, the root–shoot ratio of mycorrhizal maize decreased while the proportion of spike weight increased (Figure 6). These changes favored biomass allocation to the aboveground parts and grains. Those findings may provide insights into the utilization of AMF to reduce fertilizer use under different fertilizer regimes. 

Furthermore, the differential impacts of fertilizer reduction and phosphorus supplementation on the establishment of mycorrhizal symbiosis and mycorrhizal effects in both chemical and mixed fertilizer regimes may also be attributed to variations in soil nitrogen phosphorus interactions [60], microbial activities [61], root signals and interactions [62]. Therefore, further research is needed on underlying mechanisms and developing effective AMF strategies to tackle the globally prevalent issue of excessive agricultural fertilizer utilization.

## 5. Conclusions

In this study, we examined the impact of arbuscular mycorrhizal fungi on maize yield across various reduced fertilization regimes in medium–low fertility croplands. Our findings indicated that the mycorrhizal effect of AMF was influenced by a combination of factors including the level and type of fertilization, as well as phosphorus supplementation. Notably, reducing fertilizer levels facilitated the establishment of symbiotic mycorrhizal associations. Moreover, inoculation with AMF inoculants further enhanced root colonization and soil mycelium density. In our study area characterized by low phosphorus content, supplementing phosphorus after fertilization reduction emerged as a viable method for sustaining or increasing maize yields. Additionally, phosphorus supplementation facilitated the expression of mycorrhizal effects by AMF. Compared to conventional chemical fertilizers, the combination of phosphorus supplementation and AMF inoculation within a mixed fertilizer reduction exhibited superior maize yields and fertilizer utilization efficiency. Our study suggests that inoculating AMF is an effective approach to addressing excessive fertilizer application in farmland, but its effectiveness is closely related to agricultural fertilization practices.

## Figures and Tables

**Figure 1 microorganisms-12-01550-f001:**
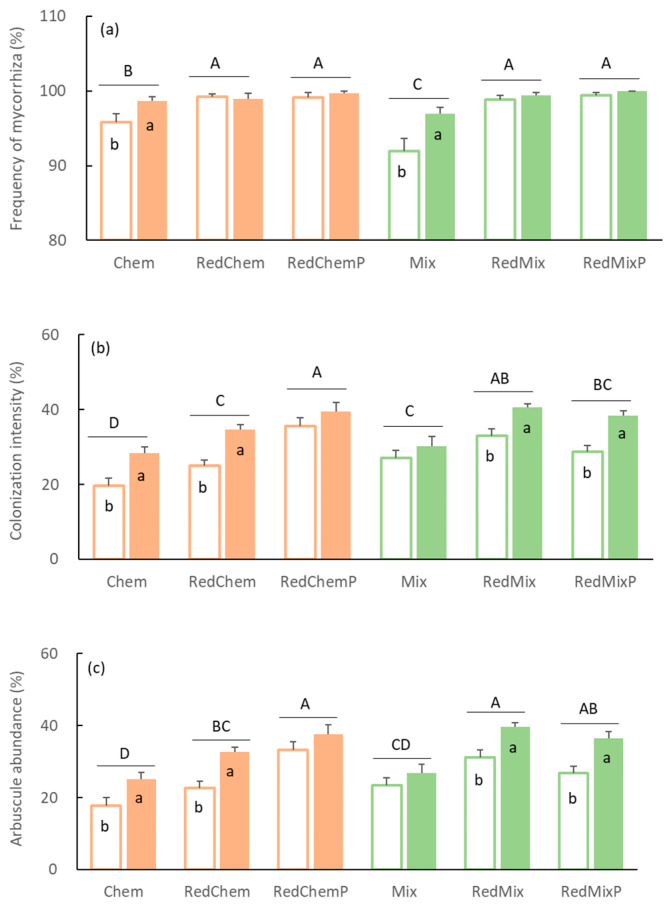
The frequency of mycorrhiza (**a**), colonization intensity (**b**), and arbuscule abundance (**c**) of maize roots in different fertilization treatments (Chem, Mix: local traditional application with chemical fertilizer or mixed fertilizer. RedChem, RedMix: 30% reduced application with chemical fertilizer or mixed fertilizer. RedChemP, RedMixP: phosphorus supplementation after 30% reduction of chemical fertilizer or mixed fertilizer) and inoculation treatments (Blank bars: without AMF inoculation. Solid bars: with AMF inoculation). Different uppercase letters above the bars represent significant differences among different fertilization treatments, while different lowercase letters on the bars represent significant differences between different inoculation treatments (*p* < 0.05).

**Figure 2 microorganisms-12-01550-f002:**
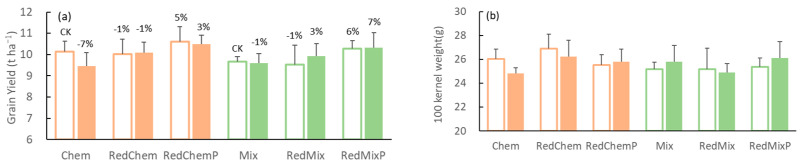
The maize grain yield (**a**) and 100 kernel weight (**b**) in different fertilization (Chem, Mix: local traditional application with chemical fertilizer or mixed fertilizer. RedChem, RedMix: 30% reduced application with chemical fertilizer or mixed fertilizer. RedChemP, RedMixP: phosphorus supplementation after 30% reduction of chemical fertilizer or mixed fertilizer) and inoculation treatments (Blank bars: without AMF inoculation. Solid bars: with AMF inoculation). The numbers above the bars represent the difference in yield compared to the yield of Chem treatment (CK, traditional fertilizer application).

**Figure 3 microorganisms-12-01550-f003:**
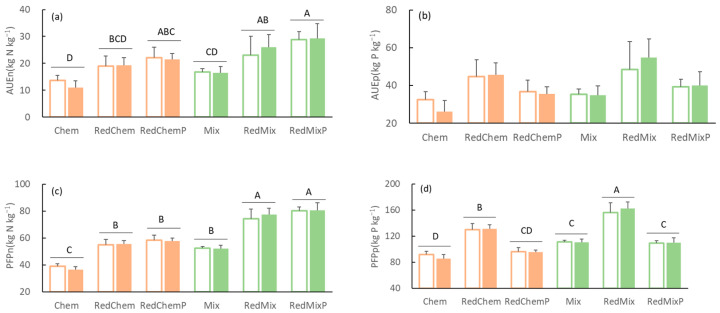
The nitrogen agronomic efficiency (AUEn, (**a**)) and phosphorus agronomic efficiency (AUEp, (**b**)), nitrogen partial factor productivity (PFPn, (**c**)) and phosphorus partial factor productivity (PFPp, (**d**)) in different fertilization (Chem, Mix: local traditional application with chemical fertilizer or mixed fertilizer. RedChem, RedMix: 30% reduced application with chemical fertilizer or mixed fertilizer. RedChemP, RedMixP: phosphorus supplementation after 30% reduction of chemical fertilizer or mixed fertilizer) and inoculation treatments (Blank bars: without AMF inoculation. Solid bars: with AMF inoculation). Different uppercase letters above the bars represent significant differences among different fertilization treatments.

**Figure 4 microorganisms-12-01550-f004:**
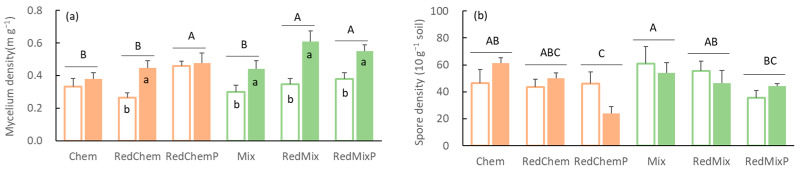
The mycelium density (**a**) and spore density (**b**) in different fertilization (Chem, Mix: local traditional application with chemical fertilizer or mixed fertilizer. RedChem, RedMix: 30% reduced application with chemical fertilizer or mixed fertilizer. RedChemP, RedMixP: phosphorus supplementation after 30% reduction of chemical fertilizer or mixed fertilizer) and inoculation treatments (Blank bars: without AMF inoculation. Solid bars: with AMF inoculation). Different uppercase letters above the bars represent significant differences among different fertilization treatments, while different lowercase letters on the bars represent significant differences between different inoculation treatments (*p* < 0.05).

**Figure 5 microorganisms-12-01550-f005:**
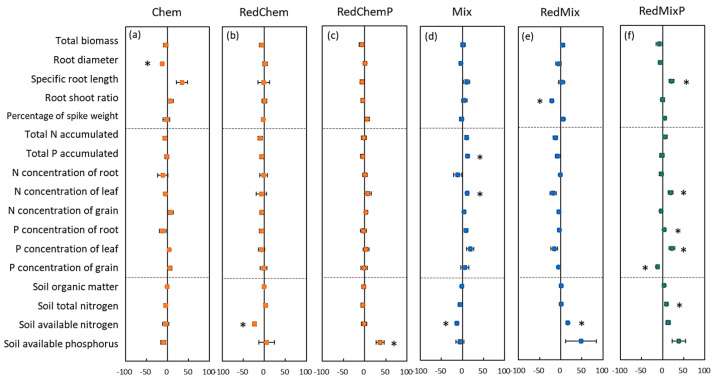
The mycorrhizal effects on plant morphological and nutrient traits, and soil nutrient contents in different fertilization treatments (Chem, Mix: local traditional application with chemical fertilizer (**a**) or mixed fertilizer (**d**). RedChem, RedMix: 30% reduced application with chemical fertilizer (**b**) or mixed fertilizer (**e**). RedChemP, RedMixP: phosphorus supplementation after 30% reduction of chemical fertilizer (**c**) or mixed fertilizer (**f**)). The asterisk represents significant mycorrhizal effect at *p* < 0.05.

**Figure 6 microorganisms-12-01550-f006:**
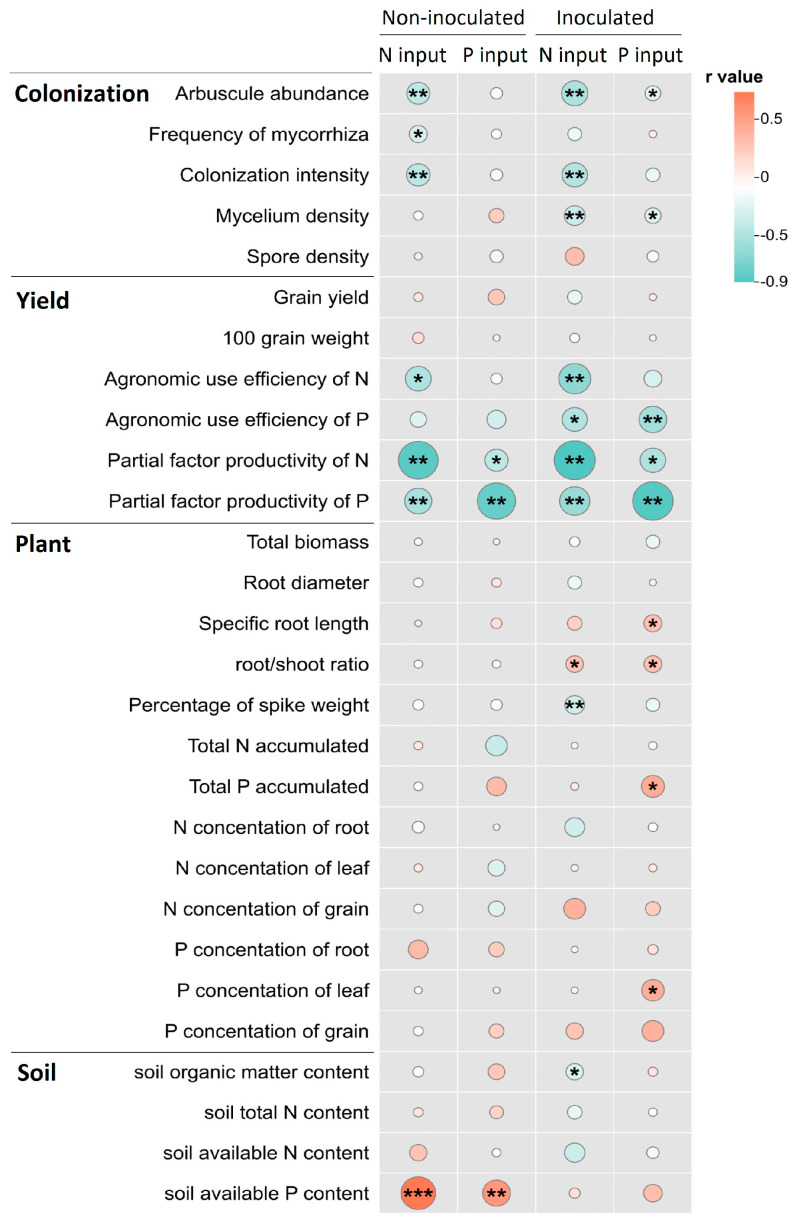
The correlation of nitrogen and phosphorus input with various variables under inoculated and non-inoculated treatments. *, **, and *** represents significant correlations at *p* < 0.05, *p* < 0.01, and *p* < 0.001.

**Table 1 microorganisms-12-01550-t001:** Types and amounts of fertilizer used for six different fertilization treatments. Each fertilization treatment included with and without AMF inoculation.

Fertilization Type	Fertilizer Treatment	Chemical Fertilizer Input (kg ha^−1^)	Organic Fertilizer Input (kg ha^−1^)	Superphosphate Input (kg ha^−1^)	Total N Input (kg ha^−1^)	N Input Reduction Ratio (%)	Total P_2_O_5_ Input (kg ha^−1^)	P Input Reduction Ratio (%)
Chemical fertilizer	Chem	1000	0	0	260	-	110	-
	RedChem	700	0	0	182	−30	77	−30
	RedChemP	700	0	208	182	−30	110	-
Mixed fertilizer	Mix	667	333	0	184	-	87	-
	RedMix	467	233	0	128	−30	61	−30
	RedMixP	467	233	208	128	−30	94	+8

The control treatment (without fertilizer and inoculation) is not shown in this table. Chem, Mix: local traditional application with chemical fertilizer or mixed fertilizer. RedChem, RedMix: 30% reduced application with chemical fertilizer or mixed fertilizer. RedChemP, RedMixP: phosphorus supplementation after 30% reduction of chemical fertilizer or mixed fertilizer.

**Table 2 microorganisms-12-01550-t002:** Two-way analysis of variance (ANOVA) using linear mixed models.

Variables	Fertilization	Inoculation	Fert. × Inocu.
Frequency of mycorrhiza	14.064_(5,130)_ ***	12.274_(1,130)_ ***	3.395_(5,130)_ **
Colonization intensity	15.374_(5,130)_ ***	43.615_(1,130)_ ***	1.140_(5,130)_
Arbuscule abundance	15.550_(5,130)_ ***	38.121_(1,130)_ ***	0.926_(5,130)_
Grain yield	0.773_(5,36)_	0.03_(1,36)_	0.184_(5,36)_
100 kernel weight	0.437_(5,36)_	0.02_(1,36)_	0.242_(5,36)_
Agronomic use efficiency of nitrogen	5.027_(5,36)_ **	0.000_(1,36)_	0.120_(5,36)_
Agronomic use efficiency of phosphorus	2.327_(5,36)_	0.000_(1,36)_	0.151_(5,36)_
Partial factor productivity of nitrogen	36.102_(5,36)_ ***	0.000_(1,36)_	0.120_(5,36)_
Partial factor productivity of phosphorus	24.482_(5,36)_ ***	0.000_(1,36)_	0.151_(5,36)_
Mycelium density	3.609_(5,132)_ **	27.884_(1,132)_ ***	2.039_(5,132)_
Spore density	2.653_(5,36)_ *	0.094_(1,36)_	1.708_(5,36)_

Data are shown as F-value with the degree of freedom as a subscript. The numbers in parentheses refer to the degrees of freedom in the numerator and denominator, respectively. *, **, and *** represents significant at *p* < 0.05, *p* < 0.01, and *p* < 0.001, respectively.

## Data Availability

Data will be made available on request.

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
