# Peer review of "Effect of AMF Inoculation on Reducing Excessive Fertilizer Use"

_microorganisms, 2024, doi:10.3390/microorganisms12081550_

Round 1

Reviewer 1 Report (Previous Reviewer 2)

Comments and Suggestions for Authors

the manuscript was improved accordingly to suggestions.,

the manuscript present useful results from maize cultivation  in  the corn belt of China., but it can be applied to several countries.

Minor details needed: 

-Line144: replace 1 by one,

-Table2: check dependent viables? or variables

- figure 6: significant relations? or correlations?

 - check for all abreviations along the text., such as AMF

-could the hypothesis   The effect of AMF on maize yield 

at various reduced fertilization regimes in medium-low fertility croplands 

 increase with reduced fertilization, and phosphorus supplementation

--line96: The original spore density of AMF in the cropland was 23 ± 96...

this could be more explained.

--how was applied the inoculant? 

--line 118:without AMF inoculation  --please add the following:(Control) to highligth controls.

line114:--mycorrhizal fungi in reduced fertilization in agricultural fields, this study established...

improve the sentence:

-mycorrhizal fungi in reduced fertilized agricultural fields, this study established...

Author Response

Reviewer 2 Report (Previous Reviewer 1)

Comments and Suggestions for Authors

The authors have followed all my comments. Everything has been corrected. I do not have any comments. The work has been corrected in the right way.

Author Response

Thank you for reviewing our manuscript.

This manuscript is a resubmission of an earlier submission. The following is a list of the peer review reports and author responses from that submission.

Round 1

Reviewer 1 Report

Comments and Suggestions for Authors

The subject of the work is interesting and current. However, the editing is unacceptable. The authors made many factual and editorial errors. The second part of the title is unnecessary. There are no cropping systems in the study. There is only fertilization. There is no correct description of the methods of analysis. Once the values are given as oxides once as the element itself. This needs to be standardized. The notation of units is incorrect. The results chapter begins with a strange passage that has no relation to the experiment. The tables give unintelligible values in parentheses. A discussion is not a discussion. Giving item numbers from references is not a proper discussion. 29 comments are marked in the text. 

Reviewer 2 Report

Comments and Suggestions for Authors

the research article entitled <Effect of AMF Inoculation in Reducing Excessive Fertilizer Use:  The Role of Agricultural Fertilization Regimes> presents relevant data on mycorrhizal inoculation in field experiments with maize in China.

the findings are relevant in order to sustain fertilizer reduction. They tested three treatments(pure, chemical and mixed). 

the Authors found positive impacts of AMF on maize grain yield using reduced chemical fertilizers. 

. Inoculation with AMF mycorrhizae significantly increased root colonization and soil mycelium density. Greater mycorrhizal effects on plant  and soil nutrients (content) were observed in mixed fertilizer treatments compared to chemical fertilizer treatments, especially when combined with phosphorus supplementation,

--Minor details  needed are:

plot size?

--site geographic coordinates?

--meteorological station coordinates?

--table 1: legend: to explain abbreviations: Red? che?

--check for abbreviations along the text

Kg/ha? or ha-1

3. Results 234 This section may be divided by subheadings. It should provide a concise and precise 235 description of the experimental results, their interpretation, as well as the experimental 236 conclusions that can be drawn.

To deletethis:

<3. Results 234 This section may be divided by subheadings. It should provide a concise and precise 235 description of the experimental results, their interpretation, as well as the experimental conclusions that can be drawn.>  

--line20: mycorrhizal agents? what it signifies? please, check, specify: mycorrhizal propagules?

-- inoculation intensity, and arbuscular abundance were conducted following the 195 method of Trouvelot[27]---is wrong 

--replace by: colonization intensity, and arbuscular abundance were conducted following the method of Trouvelot[27]

--Line21:root inoculation? or root colonization?,please,  replace this  

--3.1. Frequency of Mycorrhiza, Inoculation Intensity and...

3.1. Frequency of Mycorrhiza, colonization Intensity and

line240: inoculation intensity, and arbuscular abundance of roots. Additionally, ...

 colonization Intensity

line244: ...quency of mycorrhizal infection in maize roots, with nearly all maize roots being infected ...

infected is not more used/ replace by colonized

line 246:mycorrhizal infection was dependent on the fertilizer dosage. / wrong/

replace by mycorrhizal colonization 

line254-inoculation intensity and arbuscular abundance 

Table 2. Two-Way analysis of variance (ANOVA) using Mixed Lenear? Models

linear?

to check table 2: inoculation?colonization?

accordingly to suggestions, thus check Fig.1:mycorrhiza (a), inoculation intensity?
